# Iatrogenic Facial Nerve Palsy Following Dermatologic Cryotherapy: A Case Report and Prognostic Insights

**DOI:** 10.3390/reports7020027

**Published:** 2024-04-18

**Authors:** Michael Unterhofer, Bernhard Wenig, Peter Stoeger, Tobias Moser

**Affiliations:** Department of Neurology, Christian-Doppler Clinic, University Hospital of Salzburg, 5020 Salzburg, Austria

**Keywords:** facial nerve palsy, cryotherapy, dermatology, prognosis, neurapraxia, iatrogenic neuropathy

## Abstract

Facial nerve palsy is most commonly idiopathic, but it can also result from infections, inflammatory and cerebrovascular disorders, tumors, and trauma. We report the case of a 68-year-old patient who developed iatrogenic facial nerve palsy subsequent to dermatological cryosurgery on the right cheek. Remarkably, a full recovery occurred within 8 weeks. Drawing upon the promising outcome of this case and the existing literature on neuropathies linked with cold application in sports injuries, we propose neurapraxia as the probable pathomechanism underlying cryotherapy-induced nerve damage.

## 1. Introduction

Peripheral facial nerve palsy, characterized by sudden hemifacial weakness, is most prevalently idiopathic, while secondary causes encompass infections, inflammatory and cerebrovascular disorders, and tumors [1]. Facial nerve damage can also occur following mechanical manipulation in the proximity of the peripheral nerve branches. Iatrogenic procedures triggering facial nerve injuries include interventions in otolaryngology and maxillofacial surgeries, accounting for less than 10% of all facial nerve neuropathies [2]. We report a case of an exceptionally rare iatrogenic peripheral facial nerve palsy resulting from dermatologic cryotherapy, with a subsequent complete recovery.

## 2. Detailed Case Description

A 68-year-old woman was brought to our institution by ambulance with suspected stroke due to newfound muscle weakness on the right side of her face, which she had noticed upon waking. The patient had a history of arterial hypertension, currently managed, ulcerative colitis in remission, and chronic kidney disease. Aside from these conditions, no other relevant diagnoses were identified. She was on medication for blood pressure control and sodium bicarbonate to slow the progression of kidney disease.

The day before admission, the patient had undergone dermatological cryotherapy for seborrheic keratoses in a conventional outpatient setting. This procedure involved the precise application of liquid nitrogen on a skin lesion located in alignment with the superior aspect of the mandibular ramus, known as the coronoid process, resulting in usual cold blisters on the right cheek (Figure 1A).

Upon admission, the neurological examination revealed right-sided weakness in the mimic muscles, affecting all ipsilateral facial nerve branches, including the frontalis, orbicularis oculi, orbicularis oris, buccinator, and platysma muscles. This resulted in a partial smoothing of the brow, incomplete eye closure, mouth drop, and flattening of the nasolabial fold (Figure 1B, left). The clinical diagnosis confirmed a peripheral neuropathy affecting all branches of the right 7th cranial nerve on the ipsilateral side (grade III according to the House–Brackmann scale). This condition was attributed to dermatologic cryotherapy performed near the main trunk of the facial nerve on the preceding day. As part of the diagnostic process, a hematologic examination showed a glomerular filtration rate (GFR) of 56 mL/min, with no alterations in inflammatory markers. Considering the apparent cause of the palsy, a lumbar puncture and cerebral CT scan were deemed unnecessary. A tapering corticosteroid treatment with methylprednisolone 100 mg was prescribed over a 10-day period. The patient was discharged with a watch glass bandage, artificial tears for eye protection, and an exercise sheet for facial rehabilitation.

At the 14-day follow-up appointment, the palsy showed no significant improvement. However, after six weeks, the patient reported a notable improvement in the mimic muscles, with residual deficits in eye closure. Remarkably, two months after symptom onset, the patient had fully recovered (Figure 1B, right).

In compliance with ethical standards, written informed consent was acquired from the patient for the publication of this report. This encompassed comprehensive details from the patients’ medical history as well as photographic material.

## 3. Discussion

Cryotherapy, also known as cryosurgery, is a widely utilized and highly effective medical procedure employed across various fields, including dermatology, orthopedics, and anesthesia [3,4]. Its versatility is evident in the use of different cryogens at varying intensities depending on the medical application. For instance, in dermatology, cryogens are commonly used to freeze and remove various skin conditions. In orthopedics, cryotherapy in the form of cool packs aims to mitigate inflammation and swelling in sports injuries [5]. However, an excessive use of local ice application for athletic injuries has been reported to cause undesired tissue damage [5]. There have been documented cases of nerve palsies in the extremities, including the peroneal nerve, the lateral femoral cutaneous nerve, and the supraclavicular nerve, causing temporary disability for athletes [6,7]. Depending on the anatomical proximity concerning nerve pathways beneath the skin and the duration of cold application, cryotherapy can harm neural structures, resulting in reversible nerve palsies [6].

Concerning the anatomical variations of the facial nerve, the site of injury can yield varied clinical outcomes. Anatomically, the extratemporal facial nerve bifurcates into two primary branches, the temporofacial and cervicofacial, near the mandibular angle. A systematic review examined 1497 cadaveric facial nerve branching patterns [8]. The findings correspond with the Davis Classification, which delineates six facial nerve branching types (Type I–VI) [9]. Variations manifest in the length of the main trunk and the extent of anastomoses between branches. Apart from Type I, early bifurcation occurs in all branches (Type II–VI), along with varying numbers of additional anastomoses. Consequently, injuring one branch may not significantly affect clinical outcomes due to neural supply preservation via anastomoses from other branches. In our scenario, we presume that the patient exhibits the Type I anatomical variation of the facial nerve, present in 13–16% of cases, characterized by a lengthier main trunk and a delayed bifurcation of the two branches. Thus, we infer that the cryogenic lesion located most laterocaudally (Figure 1A) directly damaged neural tissue, resulting in neurapraxia of the main trunk. This condition impacted all ipsilateral facial nerve branches, manifesting in hemifacial weakness. Consequently, facial nerve injuries in Types II–VI may result in less pronounced clinical manifestations due to adequate nerve supply via branching anastomoses.

Cryotherapy, as a noninvasive technique, is considered a safe procedure for dermatological indications. Various cryogens, including nitrogen gas, carbon dioxide gas, and other compressed gases, are used, with liquid nitrogen being the most common. Aside from transient pain and erythema, complications of dermatological cryotherapy may include dyspigmentation, alopecia, and depressed scars [4]. To date, there are no reported cases of clinically relevant nerve injuries in dermatological indications, based on our knowledge.

In our patient, cryotherapy using liquid nitrogen impacted the primary branch of the peripheral facial nerve, resulting in transient peripheral facial motor neuropathy. The neurological deficit endured for eight weeks, with a noticeable improvement observed after six weeks. A comparable recovery pattern was observed in an animal model investigating cryosurgery of tumors affecting the facial nerve [10]. Moreover, liquid nitrogen cryotherapy, combined with the excision of lesions near the branches of the trigeminal nerve, was linked to favorable rates of sensation restoration among 16 patients initially experiencing anesthesia or paresthesia [11]. Based on the recovery pattern of our patient and the existing literature, we propose neurapraxia as the underlying pathomechanism for nerval palsy of dermatological cryotherapy. Neurapraxia is a type of nerve injury characterized by a temporary loss of nerve function without structural damage. One plausible explanation in our patient is that exposure to cold triggered vascular changes, resulting in diminished blood flow to the nerve. This, in turn, may have caused a temporary conduction blockage and dysfunction of the 7th cranial nerve (Figure 1C). Although we can only speculate regarding the etiology of cryosurgery-associated neuropathy and whether the prescribed corticosteroids contributed to complete recovery, it is noteworthy that the efficacy of cold application in pain relief has been extensively studied. Cold application’s efficacy in pain management is attributed to various mechanisms, including the suppression of nociceptor activity, the alleviation of muscle spasms, and potentially a decrease in metabolic enzyme activity levels [12]. Furthermore, cold application has been shown to modulate pain thresholds and tolerance, likely associated with reductions in nerve conduction velocity [13]. For superficial nerve injuries, direct cryolytic damage from the temporary freezing and subsequent thawing of the neural structure, known as freeze–thaw injury, appears to be the most likely pathophysiological mechanism [14]. This type of injury is primarily initiated by axonal contraction, leading to the tearing of the myelin sheath. Axonal destruction results in myelin degeneration and subsequent damage to Schwann cells, ultimately interrupting nerve impulse transmission. Two variables are recognized to influence the subsequent regeneration of neural tissue [15]. The first variable is the distance between the cryolesion and the end organ, which is directly proportional to the rate of regeneration, estimated at 1–3 mm per day. The second variable is the duration of the cryolesion, representing the time for which the tissue is exposed to temperatures below 0 °C. This duration is inversely proportional to the time required for functional regeneration [15].

It is crucial to differentiate our case, which occurred subsequent to cryotherapy, from idiopathic facial nerve palsy, commonly known as Bell’s palsy or even as “cold palsy of the VII peripheral nerve”. Bell’s palsy is frequently linked to common viral infections, notably those causing “colds”, suggesting a viral etiology as probable cause [16].

While neurapraxia, associated with a loss of conduction without associated changes in the axonal structure, is typically reversible, more severe nerve injuries, such as axonotmesis or neurotmesis, involve varying degrees of nerve fiber damage and often result in more prolonged recovery periods or the possibility of permanent deficits (Figure 1D). The underlying causes of the common forms of iatrogenic facial nerve palsies, primarily associated with complications during surgical interventions in otorhinolaryngology, oro-maxillofacial surgery, and neurosurgery and accounting for about 7% all the facial nerve palsies [2], are axonotmesis or neurotmesis. In these cases, direct damage through surgical transection leading to axonotmesis or indirect damage via toxic neurapraxia from local anesthetic application contribute to a worse long-term outcome.

## 4. Conclusions

We report on the first case of temporary cranial nerve palsy following cryosurgery, a generally safe and cost-effective treatment option for a variety of benign and malignant dermatological disorders. Consistent with the existing literature on ice application in sports, neuropathies associated with cryotherapy exhibit a favorable prognosis and good recovery, indicating neurapraxia as the underlying mechanism. 

## Figures and Tables

**Figure 1 reports-07-00027-f001:**
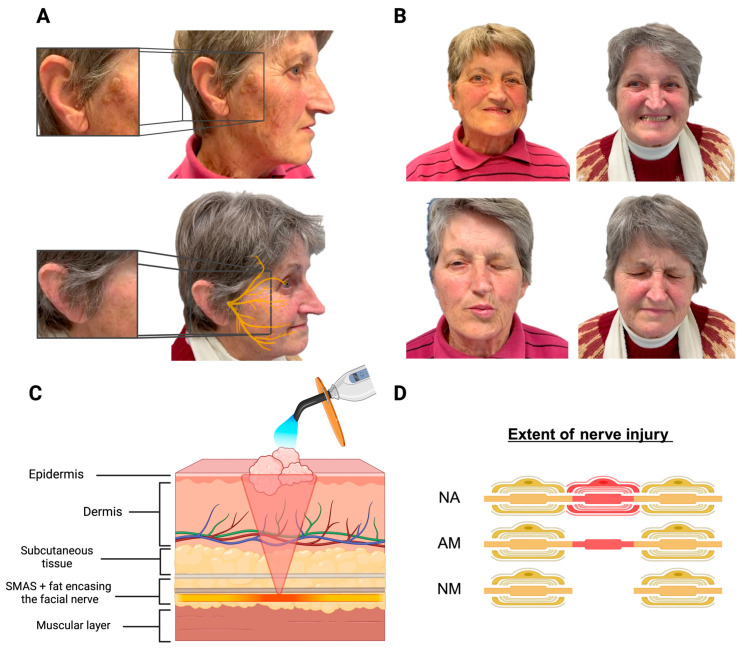
(**A**) Cold blisters on the right cheek after cryotherapy (top) and resolution after two months along with the trajectory of the facial nerve (bottom). (**B**) Left: peripheral facial nerve palsy at the time of admission; right: recovery after 2 months. (**C**) Scheme of the facial skin, facial nerve, and muscle layer; SMAS = superficial musculoaponeurotic system. (**D**) Scheme of the nerve injury classification according to Seddon; NA = Neurapraxia; AM = Axonotmesis; NM = Neurotmesis. The figure was created by biorenders.com.

## Data Availability

The original contributions presented in the study are included in the article, further inquiries can be directed to the corresponding author.

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
