# Peer review of "Iatrogenic Facial Nerve Palsy Following Dermatologic Cryotherapy: A Case Report and Prognostic Insights"

_reports, 2024, doi:10.3390/reports7020027_

Round 1
Reviewer 1 Report
Comments and Suggestions for Authors
The authors report a clinical case of a patient who presented hemifacial paralysis after a dermatological procedure. The subject is interesting for the differential diagnoses of facial paralysis.
The study is well described, presents short follow-up, pre- and post-treatment images of the patient and is described as the first case described in the literature.
Therefore, I suggest some modifications for better understanding and discussion of the clinical case.
1 - The study was sent to the ethics committee for human studies, did they obtain the patient's signature for the Consent Form? I didn't find it in the study, please add a small paragraph in the "Detailed Case Description".
2 - In image 1A, the trunk of the facial nerve is forward and above normal, in this region the temporofacial and cervicofacial division has generally already occurred. I suggest reviewing the image. Likewise, I suggest in the discussion, adding comments about the anatomical variations that the human facial nerve presents and can influence facial paralysis injuries.
3 - In the clinical case, discuss whether the paralysis was of all facial branches of the facial nerve or of the temporofacial or cervicofacial branch.
Sincerely
Comments on the Quality of English LanguageI had no difficulty reading the article.
Author Response
We would like to thank the reviewer for the time and the recommendations.
1 - The study was sent to the ethics committee for human studies, did they obtain the patient's signature for the Consent Form? I didn't find it in the study, please add a small paragraph in the "Detailed Case Description".
We added the following paragraph to the case description.
In compliance with ethical standards, written informed consent was acquired from the patient for publication of this report. This encompassed comprehensive details from the medical history as well as photographic material.
2 - In image 1A, the trunk of the facial nerve is forward and above normal, in this region the temporofacial and cervicofacial division has generally already occurred. I suggest reviewing the image. Likewise, I suggest in the discussion, adding comments about the anatomical variations that the human facial nerve presents and can influence facial paralysis injuries.
Thank you for the suggestion. We have updated the image and included a discussion on anatomical variations and their clinical implications.
Concerning the anatomical variations of the facial nerve, the site of injury can yield varied clinical outcomes. Anatomically, the extratemporal facial nerve bifurcates into two primary branches, the temporofacial and cervicofacial, near the mandibular angle. A Systematic Review examined 1497 cadaveric facial nerve branching patterns.[8] The findings correspond with the Davis Classification, which delineates six facial nerve branching types (Type I-VI).[9] Variations manifest in the length of the main trunk and the extent of anastomoses between branches. Apart from Type I, early bifurcation occurs in all branches (Type II-VI), along with varying numbers of additional anastomoses. Consequently, injuring one branch may not significantly affect clinical outcomes due to neural supply preservation via anastomoses from other branches. In our scenario, we presume the patient exhibits the Type I anatomical variation of the facial nerve, present in 13-16% of cases, characterized by a lengthier main trunk and a delayed bifurcation of the two branches. Thus, we infer that the cryogenic lesion located most laterocaudally (Figure 1A) is directly damaging neural tissue, resulting in neurapraxia of the main trunk. This condition impacts all ipsilateral facial nerve branches, manifesting in hemifacial weakness. Consequently, facial nerve injuries in Types II-VI may result in less pronounced clinical manifestations due to adequate nerve supply via branching anastomoses.
3 - In the clinical case, discuss whether the paralysis was of all facial branches of the facial nerve or of the temporofacial or cervicofacial branch.
Thank you, we have answered this concern above - please refer to the previous response.
Reviewer 2 Report
Comments and Suggestions for Authors
Well written paper. It is an unusual case presentation and therefore worthily publishing it. Especially as cryotherapy is frequently used in the elderly population and VII nerve damage fairly unusual!
Author Response
Thank you very much for your proofreading and feedback.
Reviewer 3 Report
Comments and Suggestions for Authors
Thank you for inviting me to review this submission titled “Iatrogenic Facial Nerve Palsy Following Dermatologic Cryotherapy: A Case Report and Prognostic Insights”. Here are some comments and suggestions for the authors:
- The abstract is clear and concise.
- The manuscript is well-written and well-structured.
- The illustration and the photos in Figure 1 are adequate for understanding the complete process of the injury. It would be interesting to see the evolution of the “cold blisters” as well.
- I suggest adding the following literature to the discussion section, please revise and add the corresponding information, especially regarding the pathophysiology of the nerve injury:
1. Whittaker DK. Degeneration and regeneration of nerves following cryosurgery. Br J Exp Pathol. 1974 Dec;55(6):595-600. PMID: 4447794; PMCID: PMC2072730.
2. Robert M. Beazley, Demetrius H. Bagley, Alfred S. Ketcham. The effect of cryosurgery on peripheral nerves. Journal of Surgical Research. Volume 16, Issue 3,1974.
3. Breidenbach LMThomford NPace WG. Cryosurgery of Tumors Involving the Facial Nerve. Arch Surg.1972;105(2):306–307. doi:10.1001/archsurg.1972.04180080154025
4. Brian L. Schmidt, M.A. Pogrel. Neurosensory changes after liquid nitrogen cryotherapy. Journal of Oral and Maxillofacial Surgery. Volume 62, Issue 10,2004,Pages 1183-1187.
Author Response
- The illustration and the photos in Figure 1 are adequate for understanding the complete process of the injury. It would be interesting to see the evolution of the “cold blisters” as well.
Thank you for your feedback. We have included both a before and after picture of the cold blisters.
- I suggest adding the following literature to the discussion section, please revise and add the corresponding information, especially regarding the pathophysiology of the nerve injury:
We would like to express our appreciation for your literature suggestions, as they have significantly contributed to the quality of our work. We have incorporated all four papers into our manuscript.
- Whittaker DK. Degeneration and regeneration of nerves following cryosurgery. Br J Exp Pathol. 1974 Dec;55(6):595-600. PMID: 4447794; PMCID: PMC2072730.
- Robert M. Beazley, Demetrius H. Bagley, Alfred S. Ketcham. The effect of cryosurgery on peripheral nerves. Journal of Surgical Research. Volume 16, Issue 3,1974.
For superficial nerve injuries, direct cryolytic damage from temporary freezing and subsequent thawing of the neural structure, known as freeze-thaw injury, appears the most likely pathophysiological mechanism.[14] This type of injury is primarily initiated by axonal contraction, leading to the tearing of the myelin sheath. Axonal destruction results in myelin degeneration and subsequent damage to Schwann cells, ultimately interrupting nerve impulse transmission. Two variables are recognized to influence the subsequent regeneration of neural tissue.[15] The first variable is the distance between the cryolesion and the end organ, which is directly proportional to the rate of regeneration, estimated at 1-3 millimeters per day. The second variable is the duration of the cryolesion, representing the time the tissue is exposed to temperatures below 0°C. This duration is inversely proportional to the time required for functional regeneration.[15]
- Breidenbach LM, Thomford N, Pace WG. Cryosurgery of Tumors Involving the Facial Nerve. Arch Surg.1972;105(2):306–307. doi:10.1001/archsurg.1972.04180080154025
- Brian L. Schmidt, M.A. Pogrel. Neurosensory changes after liquid nitrogen cryotherapy. Journal of Oral and Maxillofacial Surgery. Volume 62, Issue 10,2004,Pages 1183-1187.
A comparable recovery pattern was observed in an animal model investigating cryosurgery of tumors affecting the facial nerve.[10] Moreover, liquid nitrogen cryotherapy, combined with the excision of lesions near the branches of the trigeminal nerve, was linked to favorable rates of sensation restoration among 16 patients initially experiencing anesthesia or paresthesia.[11] Based on the recovery pattern of our patient and the and existing literature, we propose neurapraxia as the underlying pathomechanism for nerval palsy of dermatological cryotherapy.
Round 2
Reviewer 3 Report
Comments and Suggestions for Authors
Thank you for your response. All requirements were assessed.